# Differential Oxidative Stress Management in Industrial Hemp (IH: *Cannabis sativa* L.) for Fiber under Saline Regimes

**DOI:** 10.3390/metabo14080420

**Published:** 2024-07-31

**Authors:** Naveen Dixit

**Affiliations:** Department of Agriculture Food and Resources Sciences, University of Maryland Eastern Shore, Princess Anne, MD 21853, USA; fnaveenkumar@umes.edu

**Keywords:** salinity, reactive oxygen species, antioxidants, lipid peroxidation, hydrogen peroxide, lipoxygenase, peroxidase, catalase, glutathione reductase, glutathione-S-transferase

## Abstract

In the current study, two commercial industrial hemp (IH) fiber varieties (V1: CFX-2 and V2: Henola) were assessed for their ability to regulate salt-induced oxidative stress metabolism. For 30 days, plants were cultivated in greenhouse environments with five different salinity treatments (0, 50, 80, 100, 150, and 200 mM NaCl). Hydrogen peroxide (H_2_O_2_), malondialdehyde (MDA), and lipoxygenase (LOX) and antioxidant enzymes (superoxide dismutase (SOD), catalase, guaiacol peroxidase (GPOD), ascorbate peroxidase (APX), glutathione reductase (GR), and glutathione-S-transferase (GST)) were assessed in fully expanded leaves. At 200 and 100 mM NaCl concentrations, respectively, 30 days after saline treatment, plants in V1 and V2 did not survive. At 80 mM NaCl, the leaves of V2 showed higher concentrations of H_2_O_2_, MDA, and LOX than those of V1. Higher SOD, CAT, GPOD, APX, GR, and GST activity in the leaves of V1 up to 100 mM NaCl resulted in lower levels of H_2_O_2_ and MDA. At 80 mM NaCl, V2 demonstrated the total failure of the antioxidant defense mechanism. These results reveal that V1 demonstrated stronger salt tolerance than V2, in part due to better antioxidant metabolism.

## 1. Introduction

Soil salinity is one of the major causes of the decline in crop growth and development in changing global climatic conditions [1,2]. It has been reported that more than 20% of the cultivated land is affected by salt stress [3]. Soils are enriched with salt by both natural and anthropogenic activities, including wind, rainfall, parent rock weathering, excessive evaporation, seawater intrusion, floods, and rainfed irrigation [2]. Higher salt concentration in the root zone inhibits the growth of root and above ground plant parts by osmotic and ionic effects [4,5,6]. Moreover, the secondary effects cause the excessive production of reactive oxygen species (ROS), which are highly reactive and cause plant cell death [7]. ROS comprise superoxide radicals (O^−^), hydrogen peroxide (H_2_O_2_), hydroxyl radicals (OH^•^), and singlet oxygen (^1^O_2_) and are produced in multiple cellular organelles [7].

The successful establishment of seedlings determined the future growth and production of a plant in terms of fruit, fiber, and seed yield [1,5,6]. In fact, the potential of salt tolerance at the seedling stage has been used as a screening criterion in multiple crops [1,5,6]. Different growth stages in the life cycle of a plant showed variable sensitivity to salt stress [8]. Germination, seedling, and flowering stages are highly susceptible to salt stress in glycophytes [8,9]. Industrial hemp (IH; *Cannabis sativa* L.) for fiber is an economically important crop and grown worldwide to fulfill the natural fiber demand [9,10]. IH germplasm showed a wide range of variability against salt stress; therefore, it is imperative to explore salt-tolerant mechanisms in commercially available varieties to provide timely inputs to IH growers and plant breeders [11]. Moreover, cultivation of salt-tolerant IH for fiber will also help in reclaiming the salt-affected lands.

Little is known about the salt-induced oxidative responses of IH for fiber. Higher salt tolerance has been attributed to the higher activities of antioxidant enzymes comprising superoxide dismutase (SOD), catalase (CAT), guaiacol peroxidase (GPOD), ascorbate peroxidase (APX), glutathione reductase (GR), and glutathione-S-transferase (GST) [6,7]. Elevated levels of antioxidant enzymes have been observed in the leaves of salt-tolerant cultivars of tomatoes (GST), pistachios (SOD, GPOD, CAT), barley (APX), rice, and hemp (SOD), which showed the critical role these enzymes in the effective management of ROS [6,12,13,14,15].

Salt stress causes the production of H_2_O_2_ in leaves by the action of SOD [7]. SOD converts O^−^ into less toxic H_2_O_2_ [16]. H_2_O_2_ is a signaling molecule at low concentrations, but at higher concentrations, it promotes lipid peroxidation and plant cell death [16]. Higher leaf SOD activity is associated with a lower rate of lipid peroxidation in salt-tolerant pistachio rootstock UCB-1 [13]. In addition to ROS-mediated lipid peroxidation, salt-induced lipoxygenase (LOX) activity also promotes lipid peroxidation and membrane deterioration during senescence [17,18]. Higher levels of H_2_O_2_ in association with transition metals (Cu^+^, Fe^2+^) can generate highly reactive OH^•^ though a Fenton reaction [19]. To maintain physiological levels of H_2_O_2_, plant cells employ a battery of peroxidative enzymes, including CAT, GPOD, and APX [19]. CAT is located in peroxisomes, cytoplasm, and mitochondria and dismutates H_2_O_2_ into water and molecular oxygen without using reducing power [7]. GPOD and APX are located throughout the cell and use reducing power in the form of NADPH and a non-specific reductant, respectively [20]. Both enzymes convert H_2_O_2_ into water [7]. Higher CAT and GPOD activity have been documented in the leaves of maize genotypes in salt (NaCl: 100 mM) stress [21]. These activities were higher in the salt-tolerant genotype BR5033 in comparison to salt-sensitive BR5011 [21]. GPODs, categorized as class III peroxidases, are primarily situated in the apoplast and vacuole. These enzymes are capable of generating OH^•^ radicals, which play a crucial role in facilitating the loosening of the cell wall [22]. APX works in cooperation with GR through the ascorbate–glutathione cycle [20]. APX uses ascorbate (ASH) as an electron donor, which is generated by an intermediate enzyme, mono/dehydroascorbate reductase, by using reduced glutathione (GSH). GSH, in turn, is regenerated by the action of GR by oxidizing NAPDH [20]. GR is the last and rate-limiting enzyme in the ascorbate–glutathione cycle and essential for APX functioning [7,21]. The overexpression of APX has been shown to confer salt tolerance in transgenic plum plantlets at 100 mM NaCl and corroborated with lower levels of H_2_O_2_ in comparison to wild types [23]. Similarly, higher GR activity was evident in the high-yielding variety of *Phaseolus vulgaris* cvs. Tema in comparison to low-yielding variety Djadida in saline (NaCl: 50–200 mM) conditions [24]. All of this evidence showed that plants with well-equipped antioxidant enzymes can ensure the successful completion of the life cycle of a plant in saline regimes.

Plant GST are ubiquitous enzymes and belong to the Tau and Phi class [25]. GST plays multiple roles in plant growth and development and also protects plants from the adverse effect of salt stress [25]. The overexpression of *GsGST* from *Glycine soja* in transgenic tobacco imparts salt and drought tolerance [26]. Similarly, *GmGSTL1* from soybeans reduced salt-induced symptoms in transgenic *Arabidopsis thaliana* L. [27]. GST can scavenge salinity-induced O^−^, H_2_O_2_, OH^•^, oxidative lipid peroxide, and organic hydroperoxides and protects plants from the oxidative damage by peroxidase and transferase activity [28]. The objective of this study was to assess the salt tolerance potential and understand the antioxidant defense mechanisms in seedling leaves of commercially available varieties of IH grown for fiber production. By investigating these aspects, the research aimed to provide insights into how IH varieties can withstand salt stress and potentially reclaim salt-affected and marginal lands, particularly in coastal areas. Furthermore, the findings on managing ROS in IH could inform plant breeders on strategies to develop new salt-tolerant varieties through the targeted manipulation of antioxidant defense mechanisms. This work not only addresses immediate agricultural challenges but also lays the groundwork for future advancements in sustainable crop cultivation and land reclamation practices.

## 2. Materials and Methods

### 2.1. Plant Material and Culture

Seeds of two (V1: CFX-2 and V2: Henola) commercially available varieties of IH for fiber were purchased from Kings Agriseeds Inc., PA, USA. Seeds were tested to find out the seed germination percent in control conditions prior to experimentation, and results confirmed > 90% seed germination in all the varieties as provided on label by the company. There is no difference in the 1000 seed weight (V1: 21.5 g and V2: 20.5 g) among the tested varieties. Uniform-sized and intact seeds were selected for germination and seedling leaf studies. Seeds were sterilized with 2.5% sodium hypochlorite for 10 min, followed by extensive washing with distilled water and air drying [29].

Seeds were planted in pots (10 cm (diameter) × 8 cm (depth)) containing 60 g of sterilized vermiculite (Therm-O-Rock-East, Inc., PA, USA) and kept on plastic saucer (15.2 cm (diameter)) to provide water and nutrients from the bottom of pots. Salt treatments were provided by adding 100 mL each of 0, 50, 80, 100, 150, and 200 mM NaCl solution before planting seeds. Plants were grown for 30 days, and nutrients were provided using 1/4th strength of Hoagland solution [KNO_3_ (14 mM), Ca(NO_3_)_2_.4H_2_O (7 mM), KH_2_PO_4_ (4 mM), MgSO_4_.7H_2_O (1 mM), H_3_BO_3_ (25 µM), MnSO_4_.H_2_O (2 µM), ZnSO_4_.7H_2_O (2 µM), CuSO_4_.5H_2_O (0.5 µM), (NH_4_)_6_Mo_7_O_24_.4H_2_O (0.5 µM), and Fe-EDTA (20 µM)] on a weekly basis [30]. All the reagents were purchased from Fischer Scientific (Chicago, IL, USA). Pots were kept in greenhouse in natural light condition (700–750 µmole m^−2^ s^−1^ photosynthetically active radiation (PAR). The PAR reached 400 µmole m^−2^ s^−1^ (Spectrum Technologies, Inc., Aurora, IL, USA) on a cloudy day. Greenhouse were operated at 27 ± 2 °C temperature, 75% relative humidity, and 12 h photoperiod. Positions of pots were changed randomly to avoid positional effect. Plants were removed by washing roots with deionized water. Fully expanded leaves were processed for biochemical assays at 30 days after sowing seeds. Three independent experiments were conducted in a randomized complete block design from June to August 2022. Fifteen replicates were used for the estimation of all the biochemical parameters. All the biochemical assays were evaluated up to 100 mM NaCl in the leaves of V1 and up to 80 mM concentration in V2. Seedlings in V1 and V2 did not survive at 200 mM and 100 mM NaCl, respectively, by 30th day. Data are plotted for V1 only at 100 mM NaCl to show the elevated levels of antioxidant enzymes; while at this NaCl concentration, plants did not survive in V2 (Figure 1).

### 2.2. Assay of H_2_O_2_ Generation

Leaf samples (0.25 g) were homogenized in 0.1% trichloroacetic acid (TCA) and centrifuged (Centrifuge 5430 R, Eppendorf, Enfield, CT, USA) at 14,000× *g* for 15 min at 4 °C. The supernatant (0.3 mL) was mixed with 1.7 mL potassium phosphate buffer (pH 7.0) and 1 mL of 1 M potassium iodide (KI) solution and incubated for 5 min before measuring oxidation product at A_390_ (UV-1900, Shimadzu, Kyoto, Japan). H_2_O_2_ concentration was calculated from a standard curve prepared from known concentrations of H_2_O_2_ and expressed as μmol g^−1^ Fwt. [31].

### 2.3. Determination of Lipid Peroxidation

Lipid peroxidation was measured in terms of MDA content, a thiobarbitutric acid reactive substance (TBARS) as per Heath and Packer (1968) with slight modification [32,33]. Leaf samples (0.25 g) were homogenized in 0.1% trichloroacetic acid (TCA) and centrifuged at 12,000× *g* for 30 min. The supernatant was incubated with 20% TCA containing 0.5% thiobarbituric acid in a ratio of (1:9) for 30 min at 95 °C. This reaction was stopped by cooling the test tubes in an ice bath for 10 min, and the product was centrifuged at 10,000× *g* for 15 min. The absorbance of the supernatant was determined at 532 nm. The value for non-specific absorption at 600 nm was subtracted. The amount of MDA-TBA complex was calculated from the extinction coefficient of 155 mM^−1^ cm^−1^ and expressed as μmol g^−1^ Fwt.

### 2.4. Leaf Preparation for Enzyme Assay

Leaf samples (0.25 g) were homogenized in 100 mM potassium phosphate buffer pH 7.0 containing 0.5 mM ethylenediaminetetraacetic acid (EDTA), 0.1 mM phenylmethylsulphonyl fluoride (PMSF), and 2% polyvinylpyrrolidone (PVP) in a pre-chilled pestle and mortar. The extraction buffer also contained 5 mM ascorbate and homogenate was centrifuged at 4 °C for 30 min at 14,000× *g*. Since maintenance of consistent CAT electrophoretic mobility requires the presence of dithiothreitol (DTT), an aliquot of each sample was made at a concentration of 10 mM DTT to be used for catalase spectrophotometric assays [34]. For enzymatic analysis, supernatant was gel filtered over Sephadex G-25 (PD-10 column, Pharmacia) equilibrated with 50 mM potassium phosphate buffer pH 7.0. All enzymes activities were measured in a final volume of 3 mL using various aliquots of the supernatants (20–25 μL for SOD; 70–100 μL for CAT; 10–50 μL for GPOD; 50–100 μL APX; 70–100 μL for GR).

SOD activity was determined by the method of Dhindsa et al. [35]. A preliminary study revealed that dialysis of crude extract using cellulose ester dialysis tubing (MWCO 8000–10,000 Dalton) overnight showed no interference by small-molecular-weight impurities. Similar results were also reported in different plant species, such as peas, oats, and corn [36]. The assay solution (total volume 3.0 mL) contained 201 mM methionine, 1.72 mM NBT, appropriate aliquot of (20–25 µL) enzyme extract, 50 mM potassium phosphate buffer (pH 7.8), and 0.12 mM riboflavin. The riboflavin was added last. The tubes were shaken and placed 30 cm below a light source consisting of two 75 W fluorescent bulbs (Philips, USA). The reaction was started by switching on the light and was allowed to run for 10 min. The tubes were covered with black cloth immediately after switching off the light. Non-irradiated reaction mixtures containing the enzyme extract, which did not develop color, were used as controls. Blanks lacked enzyme in the reaction mixture and developed maximum color. Absorbance of the reaction mixture was measured at 560 nm. One unit of SOD activity was defined as the enzyme activity that caused a 50% inhibition of the initial rate of the reaction in the absence of enzyme [35]. SOD activity was expressed as units min^−1^ mg^−1^ protein.

The LOX-catalyzed reaction was followed spectrophotometrically by observing the increase in absorbance at 234 nm arising from conjugated double bonds formed during reaction. Root samples (0.5 g FW) were homogenized in 10 mL ice cooled 0.1 M potassium phosphate (pH 6.5) buffer in a pre-chilled pestle and mortar. The homogenate was allowed to stand for 10 min at 4 °C and then centrifuged at 14,000× *g* for 30 min at 4 °C. Supernatant was passed through two layers of muslin cloth to remove the fatty layer. The resulting clear supernatant was used as crude enzyme extract for the assay of LOX. The assay mixture composed of 2.975 mL substrate (0.5 mM linoleic acid) solution and 25 µL of enzyme extract. The reaction mixture was stirred rapidly, and increase in absorbance was measured at 234 nm for 20 min in 30 s intervals at room temperature [37]. LOX activity was expressed as µmole min^−1^ mg^−1^ P using extinction coefficient = 23 mM cm^−1^ for linoleic hydroperoxide.

CAT activity was assayed in a reaction mixture containing 50 mM potassium phosphate buffer (pH 7.0), 10 mM H_2_O_2_, and enzyme. The decomposition of H_2_O_2_ was followed at 240 nm using extinction coefficient = 39.4 mM cm^−1^ [38]. CAT-specific activity was expressed as µmole min^−1^ mg^−1^ protein.

GPOD activity was assayed in a reaction mixture containing 50 mM potassium phosphate buffer (pH 7.0), 10 mM H_2_O_2_, 0.05% guaiacol and enzyme. The activity was determined by the increase in absorption at 470 nm due to guaiacol oxidation by using the extinction coefficient = 26.6 mM cm^−1^ [38]. GPOD-specific activity was expressed as µmole min^−1^ mg^−1^ protein.

APX activity was assayed in a reaction mixture containing 50 mM potassium phosphate buffer (pH 7.0), 0.1 mM EDTA, 1.0 mM H_2_O_2_, 0.25 mM ascorbic acid, and enzyme. The activity was determined by the rate of ascorbate oxidation at 290 nm by using the extinction coefficient = 2.8 mM cm^−1^ [39]. APX-specific activity was expressed as µmole of ascorbate oxidized min^−1^ mg^−1^ protein.

GR activity was assayed in a reaction mixture containing 50 mM potassium phosphate buffer (pH 7.5), 3 mM DNTB (5,5-dithio-bis-2-nitrobenzoic acid), 0.1 mM EDTA, 2 mM reduced nicotinamide adenine dinucleotide phosphate (NADPH), and enzyme. The reaction was initiated by adding 0.67 mM GSSG (oxidized glutathione). The increase in absorbance at 412 nm was recorded at 25 °C for a period of 5 min [40]. GR-specific activity was expressed as nmole min^−1^ mg^−1^ protein. Protein content was determined by the method of Bradford (1976) using BSA as the standard [41].

GST activity was determined using 1-Chloro-2,4-dinitrobenzene (CDNB) as a substrate [42]. Assay buffer (0.1 M potassium phosphate (pH 6.5)) contained 5 mM reduced glutathione (GSH), 100 µL enzyme extract, and 1 mM CDNB in total volume of 3 mL. The activity was determined by increase in absorption at 340 nm using the molar extinction coefficient of the product (9.6 mM cm^−1^).

### 2.5. Statistical Analysis

Three independent experiments were conducted as a randomized complete block design with fifteen replicates for each treatment. All biochemical assays were repeated 3 times with five replicates on each seedling (*n* = 15). A two-way ANOVA was used to find differences between varieties and salt concentrations and their interactions. The means per plant were determined and subjected to analysis of variance (SAS OnDemand for Academics), which was performed and followed by Turkey’s HSD for multiple comparisons at *p* < 0.05. The standard error (SE) of the mean was also calculated.

## 3. Results

### 3.1. Hydrogen Peroxide

The levels of H_2_O_2_ significantly declined in the leaves of V1 at successive concentrations of NaCl in 30-day-old plants (Figure 2A). The minimum [134 nmole g^−1^ fresh weight (Fwt.)] levels of H_2_O_2_ were detected at a 100 mM salt concentration in the leaves of V1. In contrast, the levels of H_2_O_2_ were significantly increased in the leaves of V2 with an increase in NaCl concentrations. These H_2_O_2_ levels were 9% (50 mM) and 51% (80 mM) higher in comparison to the controls. However, H_2_O_2_ levels were 9% (50 mM), 24% (80 mM), and 29% (100 mM) lower in comparison to the controls in the leaves of V1. The H_2_O_2_ levels were 17% (50 mM) and 66% (80 mM) higher in leaves of V2 in comparison to H_2_O_2_ levels in the leaves of V1 at a similar NaCl concentration. The highest (742 nmole g^−1^ Fwt.) level of H_2_O_2_ was detected in the leaves of V2 at 80 mM NaCl concentration.

### 3.2. Lipid Peroxidation

The rate of lipid peroxidation (MDA) slightly increased in the leaves of V1 up to 80 mM and then declined at 100 mM NaCl, but these differences were non-significant (Figure 2B). However, the rate of lipid peroxidation was significantly higher in the leaves of V2 at all the tested concentrations of NaCl. The increase was 15% and 52% at 50 and 80 mM NaCl concentration, respectively, in comparison to the controls. The highest (23 nmole g^−1^ Fwt.) rate of lipid peroxidation was observed in the leaves of V2 at 80 mM NaCl. Lipid peroxidation levels were 8% (50 mM) and 48% (80 mM) higher in the leaves of V2 in comparison to the leaves in V1 at similar NaCl concentrations.

### 3.3. Superoxide Dismutase

The SOD activity consistently increased in the leaves of V1 up to 80 mM NaCl and then slightly declined at 100 mM NaCl but remained higher than the controls (Figure 2C). The increase in SOD activity was 1.5, 1.7, and 1.5 times higher at 50, 80, and 100 mM NaCl concentrations, respectively, in comparison to the control plants. The highest (5.2 units min^−1^ mg^−1^ protein) SOD activity was observed in the leaves of V1 at 80 mM NaCl. The SOD-specific activity increased up to 50 mM NaCl in the leaves of V2 and then declined at 80 mM NaCl, but this activity was higher in comparison to the controls. The SOD activity was 1.0 (50 mM) and 1.4 times (80 mM) lower in the leaves of V2 in comparison to the leaves in V1 at similar NaCl concentrations.

### 3.4. Lipoxygenase

Reciprocal trends were observed in the LOX activity among the tested varieties (Figure 3A). The LOX activity decreased with an increase in NaCl concentrations in the leaves of V1, while it increased in V2 leaves. The LOX activity declined by 1.9 (50 mM), 2.2 (80 mM), and 2.8 (100 mM) times in comparison to the controls in the leaves of V1. In the leaves of V2, LOX activity increased by 1.5 and 1.6 times in comparison to the controls at 50 and 80 mM NaCl, respectively. However, this activity was 3.0 (50 mM) and 4.0 (80 mM) times higher in the leaves of V2 in comparison to V1 leaves at similar NaCl concentrations. The lowest (11.7 nmole min^−1^ mg^−1^ protein) LOX activity was observed in the leaves of V1 at 100 mM NaCl.

### 3.5. Catalase

Catalase-specific activity increased in V1 up to a 50 mM NaCl concentration and thereafter declined at higher NaCl concentrations (Figure 3B). However, CAT activity remained higher than controls at 80 mM NaCl. In V2, CAT-specific activity decreased at 50 mM NaCl and remained constant at 80 mM NaCl. The highest (0.13 µmole min^−1^ mg^−1^ protein) CAT activity was observed at 50 mM NaCl in the leaves of V1 and the lowest (0.05 µmole min^−1^ mg^−1^ protein) observed at 100 mM NaCl. The CAT-specific activity in the leaves of V2 was 54% (50 mM) and 25% (80 mm) lower than the V1 leaves at the same NaCl concentrations.

### 3.6. Guaiacol Peroxidase

The GPOD-specific activity was significantly higher in the leaves of V2 in comparison to V1 in control conditions (Figure 3C). GPOD activity continuously increased in V1 with the advancement of NaCl concentrations. The increase was 1.1 (50 mM), 1.6 (80 mM), and 2.8 (100 mM) times higher in comparison to the controls. In the leaves of V2, GPOD-specific activity decreased with an increase in NaCl concentrations. The decrease was 1.5 (50 mM) and 1.6 (80 mm) times lower in comparison to the control leaves. Similarly, the decrease in GPOD-specific activity was 1.2 (50 mM) and 1.9 (80 mM) times lower in the leaves of V2 in comparison to the leaves of V1 at similar NaCl concentrations. The highest (0.34 µmole min^−1^ mg^−1^ protein) GPOD-specific activity was observed in the leaves of V1 at 100 mM NaCl, while the lowest (0.10 µmole min^−1^ mg^−1^ protein) was in V2 at 80 mM NaCl.

### 3.7. Ascorbate Peroxidase

Higher APX-specific activity (0.28 µmole min^−1^ mg^−1^ protein) was observed in the leaves of V1 in comparison to V2 (0.19 µmole min^−1^ mg^−1^ protein) in control regimes (Figure 4A). The APX-specific activity continuously increased in the leaves of V1 at successive NaCl concentrations up to 80 mM and then declined but remained higher than the controls. This increase in APX-specific activity was 1.2 (50 mM), 1.3 (80 mM), and 1.1 (100 mM) times higher in comparison to control leaves. In the leaves of V2, APX-specific activity increased up to 50 mM NaCl and then declined. The decline was 1.4 times lower at 80 mM NaCl in comparison to the controls. The APX-specific activity was 1.5 (50 mM) and 2.6 (80 mM) times lower in the leaves of V2 in comparison to V1 at the similar NaCl levels. The highest APX-specific activity (0.36 µmole min^−1^ mg^−1^ protein) was observed in the leaves of V1 at 80 mM NaCl. The lowest APX-specific activity (0.14 µmole min^−1^ mg^−1^ protein) was observed in the leaves of V2 at 80 mM NaCl.

### 3.8. Glutathione Reductase

The GR-specific activity significantly increased in the leaves of V1 with a successive increase in NaCl concentration (Figure 4B). The GR-specific activity was 7.3 nmole min^−^^1^ mg^−^^1^ protein in the leaves of V1 in control conditions and reached 25.6 nmole min^−^^1^ mg^−^^1^ protein at 100 mM NaCl. In the leaves of V2, GR-specific activity increased at 50 mM NaCl onward and reached 7.9 nmole min^−^^1^ mg^−^^1^ protein at 80 mM NaCl. The increase was 1.4 and 4.2 times higher at 50 and 80 mM NaCl, respectively. Similarly, the GR-specific activity was 2.1 (50 mM) and 2.8 (80 mM) times lower in the leaves of V2 in comparison to V1 at the same NaCl concentrations.

### 3.9. Glutathione-S-Transferase

The GST activity was higher (34.8 nmole min^−1^ mg^−1^ protein) in the leaves of V2 in comparison to V1 (30.3 nmole min^−1^ mg^−1^ protein) in control regimes (Figure 4C). A significant increase in GST-specific activity was evident in the leaves of V1 up to 80 mM NaCl and then declined (1.3 times) less than the level of the controls at 100 mM NaCl. The increase in GST activity was 12% and 42% at 50 and 80 mM NaCl, respectively, in comparison to the controls. In the leaves of V2, GST-specific activity significantly declined with an increase in NaCl concentrations and reached the lowest level of 9.5 nmole min^−1^ mg^−1^ protein at 80 mM NaCl. Similarly, the decrease in GST-specific was 2 times (50 mM) and 5.5 times (80 mM) lower in the leaves of V2 in comparison to V1 at similar NaCl concentrations. The highest (52 nmole min^−1^ mg^−1^ protein) GST-specific activity was observed in the leaves of V1 at 80 mM NaCl.

## 4. Discussion

Salinity is detrimental to plant growth and development and imparts adverse effects by modifying osmotic potential and ion concentration in different plant parts [7,16]. Lower water/osmotic potential and toxic ion concentrations disrupt normal metabolism in plant cells and generate excessive amounts of deleterious ROS [5,43]. A higher accumulation of H_2_O_2_ was evident in the leaves of V2 with an increase in NaCl concentrations (Figure 2A). The highest (742 nmole g^−1^ Fwt.) level of H_2_O_2_ was detected in the leaves of V2 at 80 mM NaCl. However, the concentration of H_2_O_2_ declined in V1 and reached a lower value of 134 nmole g^−1^ Fwt. at 100 mM NaCl. The decline was 2.5 times lower in comparison to the controls. Salinity induces physiological drought by decreasing the water potential in the root zone, thus causing water deficiency-induced stomatal closure and the non-availability of carbon dioxide for photosynthesis [9,44,45]. In these conditions, electrons leak from chloroplastic electron transport chain to molecular oxygen and generate O·^−^ [7,46]. Furthermore, exposure to salt stress can lead to alterations in both the morphology and anatomy of chloroplasts, characterized by the accumulation of starch and lipids, which subsequently results in the disintegration of grana [47]. These physiological changes can contribute to a decrease in chlorophyll a fluorescence parameters (F0; minimum fluorescence, Fm: maximum fluorescence, Fv: variable fluorescence) in hemp, which indicates photoinhibition and collapse of photochemistry in chloroplasts [9]. The decline in Fv/Fm (quantum yield of photosystem II) and Fv/F0 (activity of water splitting complex) and increase in F0 at 4 and 6 dSm^−1^ in hemp showed photoinhibition and impaired the functioning of photochemical apparatus [9,46,47]. In addition, complex I and III of the mitochondrial electron transport chain also produce O·^−^ in saline conditions [48]. We anticipate that water constraints exist in NaCl-treated IH plants, as elevated levels of H_2_O_2_ also reduce root hydraulic conductivity (by 2 to 3 times at concentrations of 0.1 to 9 mM H_2_O_2_). This reduction in conductivity impairs aquaporin function, which can further limit water availability in NaCl-treated IH plants [49,50]. It is apparent that salinity impedes water movement in roots and later a secondary effect of salinity; H_2_O_2_ accumulation further reduced water permeability in plant tissues. Lower levels (1.2 times; 50 mM NaCl and 3.0 times; 80mM NaCl) of H_2_O_2_ in V1 in comparison to V2 showed that these plants are better protected from the deleterious effects of O·^−^, as SOD converts O·^−^ into H_2_O_2_. H_2_O_2_ at lower concentrations is less toxic in comparison to O·^−^ [19].

Higher SOD activity (1.0 times; 50 mM NaCl and 1.4 times; 80mM NaCl) was observed in the leaves of V1 at all the tested NaCl concentrations in comparison to V2 (Figure 2C). SOD activity also increased in the leaves of V2 at 50 mM (1.3 times) and declined at 80 mM NaCl (1.1 times) in comparison to SOD activity at 50 mM NaCl but remained higher than the controls. However, the increase in SOD activity was lower (1.3 times; 50 mM NaCl and 1.1 times; 80mM NaCl) in V2 in comparison to V1 (1.5 times; 50 mM NaCl and 1.7 times; 80 mM NaCl). Two different kinds of situations exist in V1 and V2 in relation to specific SOD activity and H_2_O_2_ production: I: An increase in SOD activity in V1 is accompanied by the lower levels of H_2_O_2_, which can be explained on the basis of higher specific activities of CAT, POD, and APX in the leaves of V1 (Figure 3B,C and Figure 4A). All these enzymes cooperatively regulate the physiological levels of H_2_O_2_ in the plant cells [7]. II: In spite of lower increase in SOD activity in comparison to V1, higher levels (742 nmole g^−1^ Fwt.) of H_2_O_2_ were detected in the leaves of V2 due to lower CAT-, POD-, and APX-specific activities (Figure 3B,C and Figure 4A). These data indicate that V1 and V2 both showed SOD protection, but it was higher in V1. The ectopic overexpression of *TaSOD5* from wheat in *A. thaliana* L. enhances SOD activity and confers salt tolerance (200 mM NaCl) and resistance against oxidative stress (10 mM H_2_O_2_) [51]. SOD declined in both V1 and V2 at 100 mM and 80 mM NaCl, respectively, but levels were higher than the controls. There are reports that documented that higher salt (Na^+^, Cl^−^) and H_2_O_2_ concentrations inactivate SOD, which is evident in the current work [52,53]. A moderate increase in SOD activity in V2 at 50 and 80 mM NaCl compromised the first line of defense against O^−^, and simultaneously, higher H_2_O_2_ in association with Cu^+^ and Fe^2+^ can generate highly reactive OH^•^ through a Haber-Weiss/Fenton reaction [19].

ROS, including O^−^, H_2_O_2_, and OH^•^, initiate lipid peroxidation, protein carbonylation, and DNA damage [19]. Higher H_2_O_2_ levels (742 nmole g^−1^ Fwt.; 80 mM NaCl) were corroborated with the higher rate of lipid peroxidation (23.3 nmole g^−1^ Fwt.; 80 mM NaCl) in terms of malondialdehyde (MDA) content in the leaves of V2 (Figure 2B). Similarly, lower levels of H_2_O_2_ (155 nmole g^−1^ Fwt.; 80 mm NaCl) corresponded to the lower levels (12.1 nmole g^−1^ Fwt.; 80 mm NaCl) of MDA in the leaves of V1. These data clearly showed salinity-induced H_2_O_2_ production with a corresponding increase in the MDA levels in the salt-sensitive IH variety V2, while V1 maintained lower levels of H_2_O_2_ and MDA and exhibited salt tolerance. MDA is a reactive organic molecule, and its consistent higher accumulation causes cell death, which is evident in the leaves of V2 [54]. MDA is produced in cells non-enzymatically through ROS and enzymatically by the action of LOX [55]. LOX activity consistently increased (1.5 and 1.6 times at 50 and 80 mM NaCl, respectively) in V2 and had the highest activity (53.7 nmole min^−1^ mg^−1^ protein) at 80 mM NaCl (Figure 3A). However, LOX activity declined (14.8 nmole min^−1^ mg^−1^ protein: 80 mM NaCl) in V1 with an increase in NaCl concentrations. It is clear that MDA is contributed by both LOX and H_2_O_2_ in salt-stressed IH. However, it is difficult to ascertain the contribution of each component with the current data. Lower LOX activity in V1 is responsible for the lower levels of MDA and confer salt tolerance in comparison to V2 with higher LOX activity. LOXs are non-heme iron-containing dioxygenase that catalyze the incorporation of oxygen into polyunsaturated fatty acids and generate 9- and 13-hydroperoxyl fatty acids [56]. Elevated levels of LOX activity were observed in salt-stressed wheat seedlings, which are accompanied by the increased expression of *TdLpx-A2* and higher levels of MDA and O·^−^ [41]. In rice, salt-sensitive cultivars displayed higher LOX and MDA levels, while LOX remained unchanged with lower levels of MDA in salt-tolerant cultivars [57]. A similar condition was observed in salt-sensitive IH variety V2.

H_2_O_2_ in plant cells is metabolized by a machinery of peroxidative enzymes, including CAT, GPOD, and APX, located in multiple cellular locations [19]. CAT levels increased (1.3 times) in V1 up to 80 mM and then declined (1.2 times) at 100 mM NaCl. However, CAT declined (1.5 times) in V2 at 50 mM onwards and remained low, which explains the higher accumulation of H_2_O_2_ in these leaves (Figure 3B). It has been reported that 10 µM H_2_O_2_ inhibits the carbon dioxide fixation in isolated chloroplast by 50% [58]. Leaves in V1 showed lower levels of H_2_O_2_ due to highly active CAT that protects photosynthetic machinery from the adverse effects of H_2_O_2_. CAT removes the bulk of H_2_O_2_ from the system due its low *K*_M_ (40–600 mM) and high affinity for H_2_O_2_ [59]. A decline in CAT-specific activity at 80 mM and higher NaCl concentrations showed its sensitivity for NaCl toxicity [60]. Transgenic Chinese cabbage plants with maize *CAT* and *Cu/Zn-SOD* genes showed higher salt (200 mM NaCl) tolerance with elevated levels of K^+^, Ca^2+^, and Mg^2+^ and less Na^+^ in the leaves in comparison to controlled plants [61]. Higher SOD and CAT activity up to 50 mM NaCl ensures lower levels of H_2_O_2_ in V1. However, this combined protective mechanism is absent in the leaves of V2. APX has low *K_M_* and high affinity for H_2_O_2_; therefore, it is responsible for maintaining the physiological levels of H_2_O_2_ in plant cells [7]. APX utilizes reduced ascorbate to convert H_2_O_2_ into water [7]. The levels of APX increased (1.2, 1.3, and 1.1 times at 50, 80, and 100 mM NaCl, respectively) and remained higher in the leaves of V1 up to 100 mM NaCl. In comparison, APX-specific activity only increased (1.1 times) up to 50 mM NaCl and thereafter declined in V2 (Figure 4A). APX fine-tunes the levels of H_2_O_2_ and higher APX in association with CAT, and GPOD suppressed the levels of H_2_O_2_ (134 nmole g^−1^ Fwt.) below the controls (331 nmole g^−1^ Fwt.) in V1 at 100 mM NaCl. APX protection at 50 mM NaCl in V2 was not sufficient to lower the levels of H_2_O_2_ (363 nmole g^−1^ Fwt.) when CAT (0.06 µmole min^−1^ mg^−1^ protein) and GPOD (0.11 µmole min^−1^ mg^−1^ protein) were operating well below the levels of the control (CAT; 0.09 µmole min^−1^ mg^−1^ protein, GPOD; 0.16 µmole min^−1^ mg^−1^ protein). APX declined at 80 mM NaCl in V2 because of an excessive buildup of H_2_O_2_. Higher H_2_O_2_ inactivates APX at lower reduced ascorbate (ASH) concentrations, which generally exist in oxidative stress conditions [62].

GR and APX operate in conjunction through the Halliwell–Asada pathway to metabolize H_2_O_2_ using cellular redox in the form of ASH, GSH (reduced glutathione), and NADPH [20]. GR assists in the regeneration of ASH from NADPH for the functioning of APX via an intermediate enzyme mono/dehydroascorbate reductase [20]. GR-specific activity continuously increased (1.8, 3.1, and 3.5 times at 50, 80, and 100 mM NaCl, respectively) in V1 leaves up to 100 mM NaCl (Figure 4B) and supports the functioning of APX. GR levels also increased in the leaves of V2, but the increase was lower (1.4 and 1.8 times at 50 and 80 mM NaCl, respectively) than V1 at the same NaCl concentrations, which showed a less active Halliwell–Asada pathway in V2. Moreover, APX declined at 80 mM NaCl in V2 when the levels of GR were high, thus causing the complete failure of the Halliwell–Asada pathway. Higher GR activity in V1 in comparison to V2 can further impart salt tolerance by producing GSH, which can directly scavenge OH^•^ and ^1^O_2_ [63]. In addition, GSH also maintains a reducing environment in the cell, which is essential for the function of proteins [64]. Several authors suggested that higher GR activity is associated with salt tolerance in salt-tolerant cultivars of pea and mulberry [65,66].

GSTs also required GSH for the scavenging of ROS and lipid peroxides [27]. GST activity increased [12% (50 mM NaCl) and 42% (80 mM NaCl)] in the leaves of V1 up to 80 mM NaCl and then declined (1.3 times) at 100 mM NaCl in comparison to the controls (Figure 4C). However, GST declined in V2 at all the tested concentrations of NaCl and reached a lower value of 9.5 nmole min^−1^ mg^−1^ protein at 80 mM NaCl, which is 3.7 times lower than the controls. The GST-mediated antioxidant defense is operative in V1 up to 80 mM NaCl; however, V2 leaves are devoid of this defense mechanism. Recently, GSTU7 has been found in the cytoplasm of *A. thaliana* L., which acted as peroxidase and contributed to oxidative resistance in cells [27]. Multiple reports showed that the overexpression of the GST gene enhances salt tolerance in transgenic plants [26,67]. Higher GST levels in V1 confer salt tolerance in association with the other components (CAT, GPOD, APX, and GR) of antioxidant defense systems and coincided with the lower levels of MDA and H_2_O_2_. Apparently, V2 showed a poor performance of the antioxidant defense system (CAT, GPOD, APX, GST), which is accompanied by higher concentrations of MDA and H_2_O_2,_ thus is not a suitable variety for salt-affected soils. This study demonstrated the substantial role of enzymatic antioxidant metabolism in IH for fiber in saline regimens. Higher activities of SOD, CAT, GPOD, APX, GR, and GST in V1 maintained the lower levels of H_2_O_2_ and MDA with concomitant plant survival up to 100 mM NaCl. Plant breeders and biotechnologists can enhance the enzymatic antioxidant metabolism of fiber hemp to develop salt-tolerant varieties, leveraging our research’s foundational insights into ROS management under saline conditions. Moreover, fiber hemp can be utilized by marginal farmers and those in coastal areas for land reclamation purposes.

## 5. Conclusions

IH for fiber is an economically important crop worldwide and can be grown in marginal and salt-enriched soils. In the current work, two commercially available IH varieties for fiber showed differential responses for salt stress tolerance and corresponding oxidative stress metabolism in leaves. Salt tolerance in V1 is mainly governed by higher levels of antioxidant enzymes with a concomitant decline in H_2_O_2_ and MDA. However, V2 leaves failed to maintain lower levels of H_2_O_2_ and MDA. These data showed the existence of variations in salt tolerance in IH for fiber, which is mediated by enzymatic components of antioxidant metabolism, and can be utilized for future salt management breeding programs in IH.

## Figures and Tables

**Figure 1 metabolites-14-00420-f001:**
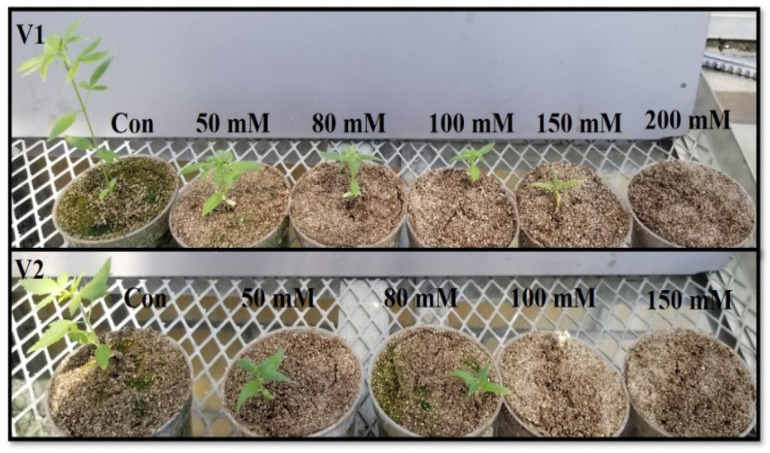
The effects of NaCl on IH leaves at 0, 50, 80, and 100 mM in V1 and 0, 50, and 80 mM in V2. Seedlings in V1 and V2 did not survive at 200 mM and 100 mM NaCl, respectively, by the 30th day.

**Figure 2 metabolites-14-00420-f002:**
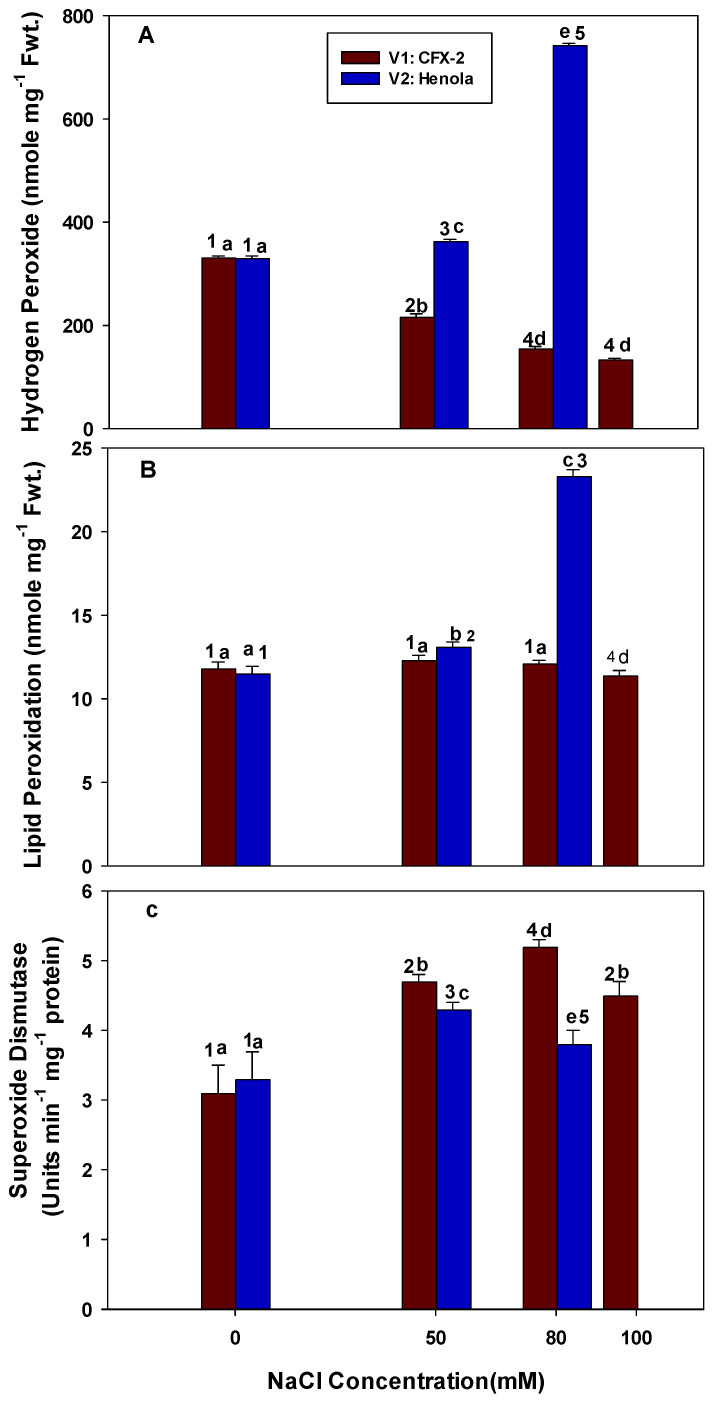
The effects of NaCl on the IH leaf H_2_O_2_ concentration (Panel (**A**)), lipid peroxidation (Panel (**B**)), and SOD-specific activity (Panel (**C**)). Vertical bars represent SE. Different lowercase letters indicates significant differences among the varieties, and different numerals represents significant treatment differences. These mean values (*n* = 15) are separated using LSD at *p* < 0.05. Data at 100 mM NaCl are given for V1 only, as plants did not survive in V2 at this NaCl concentration. Interactions were non-significant.

**Figure 3 metabolites-14-00420-f003:**
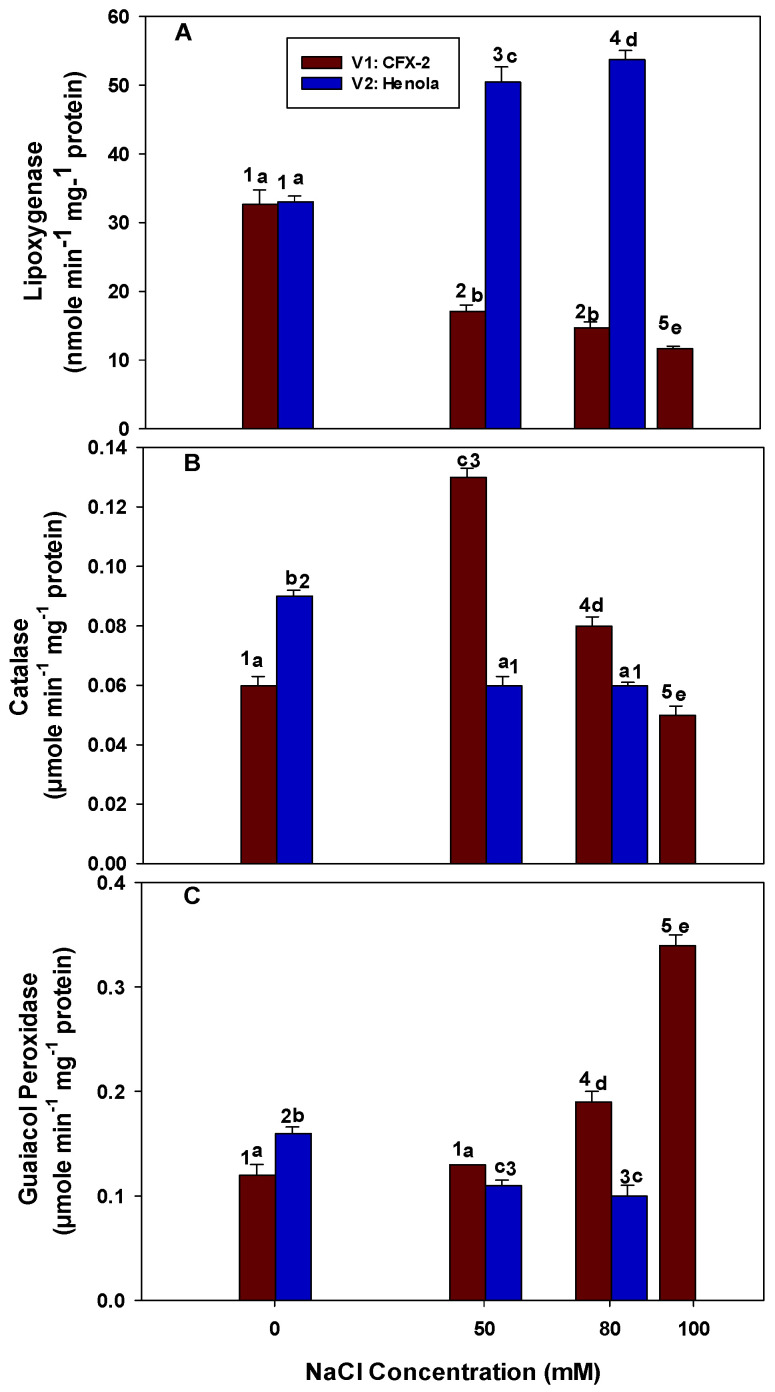
The effects of NaCl on IH leaf LOX-specific activity (Panel (**A**)), CAT-specific activity (Panel (**B**)), and GPOD-specific activity (Panel (**C**)). Vertical bars represent SE. Different lowercase letters indicates significant differences among the varieties, and different numerals represent significant treatment differences. These mean values (*n* = 15) are separated using LSD at *p* < 0.05. Data at 100 mM NaCl are given for V1 only, as plants did not survive in V2 at this NaCl concentration. Interactions were non-significant.

**Figure 4 metabolites-14-00420-f004:**
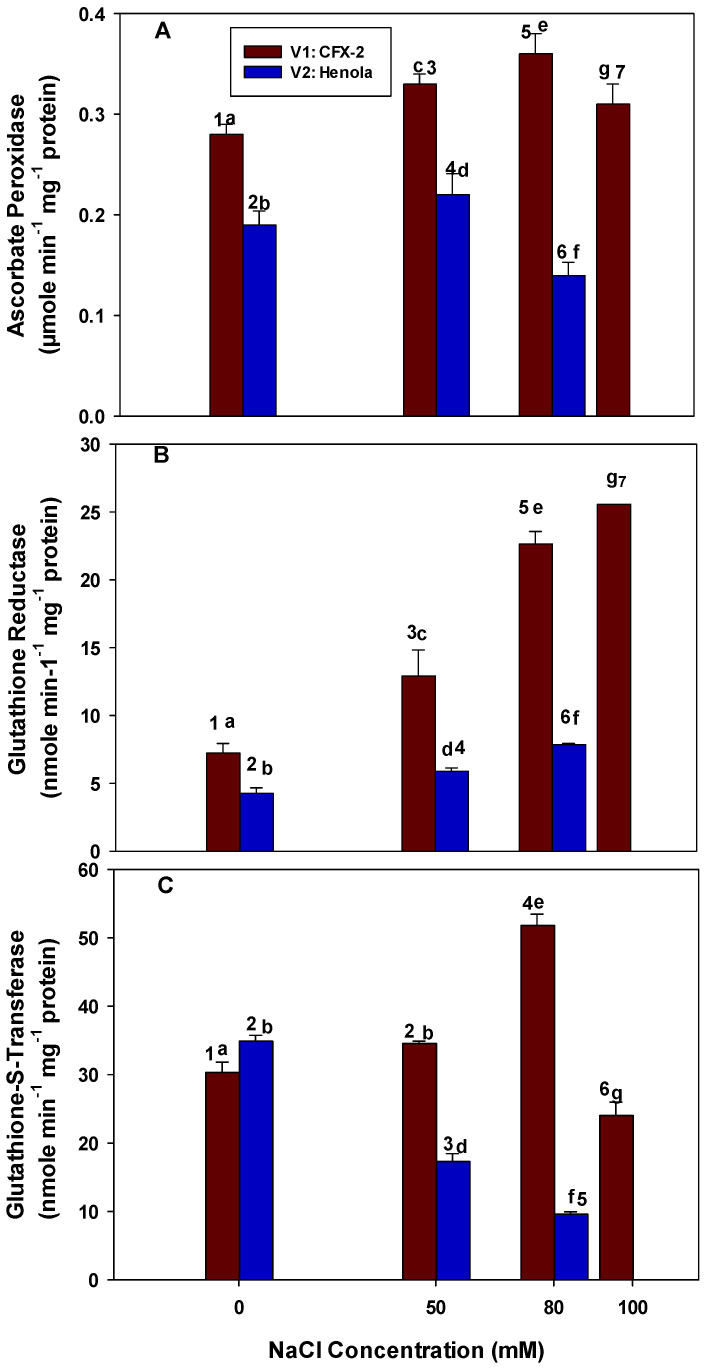
The effects of NaCl on IH leaf APX-specific activity (Panel (**A**)), GR-specific activity (Panel (**B**)), and GST-specific activity (Panel (**C**)). Vertical bars represent SE. Different lowercase letters indicate significant differences among the varieties, and different numerals represent significant treatment differences. These mean values (*n* = 15) are separated using LSD at *p* < 0.05. Data at 100 mM NaCl are given for V1 only, as plants did not survive in V2 at this NaCl concentration. Interactions were non-significant.

## Data Availability

The data presented in this study are available in the article.

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
