# Peer review of "Differential Oxidative Stress Management in Industrial Hemp (IH: Cannabis sativa L.) for Fiber under Saline Regimes"

_metabolites, 2024, doi:10.3390/metabo14080420_

Round 1

Reviewer 1 Report

Comments and Suggestions for Authors

Manuscript Oxidative Stress Metabolism in Industrial Hemp (IH: Cannabis sativa L.) for Fiber Under Sodium Chloride Salinity by

Naveen Dixit examines a wide range of physiological indicators of salt effects on the formation of Fiber Under Sodium Chloride and associated biochemical characteristics.

The manuscript is formatted according to the rules and contains the necessary sections and an extensive list of references. Statistical support corresponds to the modern level.

There are a number of things that can detract from the significance of this study and its presentation.

The title of the article does not contain a defining verb, in addition, most likely this is a question, because otherwise it turns out that the author believes that oxidative stress exists for Fiber Under Sodium Chloride Salinity, which hardly makes sense.

The objectives of the work remain unclear.

So, at the end of the introduction, namely in the last paragraph, it is customary to provide information about the purpose and objectives of the study, or better yet, about the hypothesis (hypotheses) that the author is trying to test. However, the introduction ends with a discussion of private data.

It is very sad that the author ignored those subcellular structures that are affected by oxidative stress. This is not discussed in the introduction or discussion.

I propose to expand this section using materials indicating damage to plastids, as the main source of construction and energy resource of plant cells, the cytoskeleton, features of the formation of vacuoles, which are crucial in growth by extension, the state of the cytoskeleton that ensures growth by division and extension.

It is better to place such a fragment in the discussion, in accordance with the identified inhibitory processes.

Review options on this topic:

Liu, J., Zhang, W., Long, S., & Zhao, C. (2021). Maintenance of cell wall integrity under high salinity. International Journal of Molecular Sciences, 22(6), 3260.

Zahra, N., Al Hinai, M. S., Hafeez, M. B., Rehman, A., Wahid, A., Siddique, K. H., & Farooq, M. (2022). Regulation of photosynthesis under salt stress and associated tolerance mechanisms. Plant Physiology and Biochemistry, 178, 55-69. 

Baranova, E. N., & Gulevich, A. A. (2021). Asymmetry of plant cell divisions under salt stress. Symmetry, 13(10), 1811.

Kumar, S., Jeevaraj, T., Yunus, M. H., Chakraborty, S., & Chakraborty, N. (2023). The plant cytoskeleton takes center stage in abiotic stress responses and resilience. Plant, Cell & Environment, 46(1), 5-22.

Mansour, M. M. F. (2023). Role of vacuolar membrane transport systems in plant salinity tolerance. Journal of Plant Growth Regulation, 42(3), 1364-1401.

or other.

In addition, it is not discussed how exactly the effect of salt, unfavorable for cell growth and therefore plants, occurs in terms of the studied parameters.

It is extremely important to provide photos of plants in order to understand exactly how growth was inhibited.

Regarding the design of the drawings: they should be made clearer and enlarged, the word treatments should be replaced with NaCl concentration or NaCl treatments.

Probably the article was prepared for a paper magazine and therefore contains black and white drawings; in electronic magazines this looks unjustified and can be easily changed.

What is especially striking is that neither in the discussion nor in the conclusion is there a clearly expressed thought about the impact and the mechanism of this impact specifically on the Fiber topic indicated in the title. Sometimes it is not even obvious that the measured characteristics are related. To improve perception, you can add the author’s proposed interaction scheme for the studied biochemical data or a correct and simple description.

Minor notes:

I was unable to find a definition of germination and characteristics of seedlings and their photos

It is recommended to break down the discussion into outcome points.

This manuscript is of interest to agricultural production technologists and breeders and can be published after

Reviewer 2 Report

Comments and Suggestions for Authors

This study investigated the oxidative stress metabolism in industrial hemp under salinity. These findings will be utilized for future salt management breeding programs in industrial hemp. However, there are some questions need to be answered before this manuscript can be published.

1. I do not think industrial hemp can be abbreviated to IH.

2. In the sentences 49-50, [Hasanuzzaman et a., 2021]?

3. In the Introduction, I can not see the aim of this study, or I can not find any questions will be solved in study.

4. In the sentences 92, what is the SGP meanings?

5. In the sentences 216-217, The minimum [235 nmole g-1 fresh weight (Fwt.)] levels of H2O2 were detected at 100 mM salt concentration in leaves of V1. 235 nmole g-1 fresh weight (Fwt.) is correct?

6. In the Figure, Different capital letters indicates significant differences among the varieties and different numerals represents significant treatment differences. where is the capital letters? I also can not understand this sentences, because there are both letters and numerals in the figures, so it is complex. Please revise it. The horizontal coordinate-axis X is treatments (mM) should changed into NaCl concentration (mM).

7. In the sentences 233-234, The rate of lipid peroxidation (MDA) slightly increased in the leaves of V1, but these differences were non significant. in this sentence, slightly increased is correct? MDA content under 100 mM NaCl is lower than that in control.

8. The format of the references is not accurate, for example, References 6, Huaran, H.; Hao, L.; Guanghui, D.; Fei, Y.; Gang, D.; Yang, Y.; Feihu, L. 2019. Fiber and seed type of hemp (Cannabis sativa L.) responded differently to salt-alkali stress in seedling growth and physiological indices. Ind. Crop. Prod. 2019, 129, 624-630. is not accurate. Should be Hu, H.; Liu, H.; Du, G.; ...

9. Suggest adding some pictures of industrial hemp growth after salt treatment.

Reviewer 3 Report

Comments and Suggestions for Authors

In this manuscript, the author reported the responses of two commercial industrial hemp varieties to salt stress. Biochemical parameters, i.e., hydrogen peroxide, malondialdehyde, lipoxygenase activity, and a number of antioxidant enzymes were comparatively analyzed. The author has carefully reported details of the experiments performed. Gap of knowledge regarding the responses of hemp plants to salt-induced oxidative stress was highlighted. Scope of the study seems somewhat narrow, although the manuscript was overall well presented. I have only very minor feedbacks as listed below.

Below are my feedbacks for the author’s consideration:

1.     ABSTRACT line 16: “But in V1, these amounts decreased.” – This part is unclear to me. The “decrease” was found when compared to?

2.     ABSTRACT line 17: The abbreviations “GPOD” and “APX” should be introduced when first mentioned.  

3.     Did the author have any photos that they can provide as an evidence of the differential growth performance of the two varieties investigated? In Line 349, the author mentioned Figure 4. But there is no Figure 4 anywhere in the manuscript. Is Figure 4 supposed to be a photo of the two plants growing in salt-treated vermiculate?

4.     In Line 348: “…which is evident from the decline in plant height with increases in salt concentrations…” -  Did the author mean that both V1 and V2 showed decline in plant height when salt-stressed? If so, was the differences in plant height between the two varieties statistically significant?

Round 2

Reviewer 1 Report

Comments and Suggestions for Authors

The manuscript Oxidative Stress Metabolism in Industrial Hemp (IH: Cannabis sativa L.) for Fiber Under Sodium Chloride Salinity by Naveen Dixit has been significantly improved, with most of the comments removed.

Taking into account the changes made, the manuscript can be published

Reviewer 2 Report

Comments and Suggestions for Authors

Thank you for your modified version, but there are also some problem to be revised.

1. I do not think industrial hemp can be abbreviated to IH, so I suggest to delete it.

2. Different lowercase letters indicates significant differences among the varieties and different numerals represents significant treatment differences. I also think that it was very complex, so only use the lowercase letters to indicate significant differences among the varieties and treatment (all the columns).

3. P<0.05, P should be italic.
